# Giant ferroelectric polarization in a bilayer graphene heterostructure

Ruirui Niu[1], Zhuoxian Li[1], Xiangyan Han[1], Zhuangzhuang Qu[1], Dongdong Ding[1], Zhiyu Wang[1], Qianling Liu[1], Tianyao Liu[1], Chunrui Han [2,3], Kenji Watanabe [4], Takashi Taniguchi [4], Menghao Wu[5], Qi Ren[6], Xueyun Wang [6], Jiawang Hong [6], Jinhai Mao [7], Zheng Han [8,9], Kaihui Liu [1], Zizhao Gan[1] & Jianming Lu [1] ✉

At the interface of van der Waals heterostructures, the crystal symmetry and the electronic structure can be reconstructed, giving rise to physical properties superior to or absent in parent materials. Here by studying a Bernal bilayer graphene moiré superlattice encapsulated by 30°-twisted boron nitride flakes, we report an unprecedented ferroelectric polarization with the areal charge density up to $10^{13}\,cm^{-2}$, which is far beyond the capacity of a moiré band. The translated polarization ~5 pC m$^{-1}$ is among the highest interfacial ferroelectrics engineered by artificially stacking van der Waals crystals. The gate-specific ferroelectricity and co-occurring anomalous screening are further visualized via Landau levels, and remain robust for Fermi surfaces outside moiré bands, confirming their independence on correlated electrons. We also find that the gate-specific resistance hysteresis loops could be turned off by the other gate, providing an additional control knob. Furthermore, the ferroelectric switching can be applied to intrinsic properties such as topological valley current. Overall, the gate-specific ferroelectricity with strongly enhanced charge polarization may encourage more explorations to optimize and enrich this novel class of ferroelectricity, and promote device applications for ferroelectric switching of various quantum phenomena.

A ferroelectric endowed with electrically switchable dipoles is promising in the application of nonvolatile electronics with fast switching speed, non-destructive readout, and capability of high-density integration[1–3]. The device miniaturization towards the two-dimensional limit, however, has long been hindered by the incomplete screening of depolarization fields at the surface[4]. Van der Waals (vdW) materials and their heterostructures with chemically inert interfaces open more opportunities to realize atomically thin

ferroelectrics[5–9]. Various types of ferroelectricity have been discovered in the exfoliated mono- or few-layer flakes from bulk polar vdW crystals[10–23], e.g., out-of-plane ferroelectricity in $CuInP_2S_6$, $d$1T-$MoTe_2$ and $Bi_2O_2Se$, in-plane ferroelectricity in SnS, SnSe, and β-$In_2Se_3$, and intercorrelated ferroelectricity in α-$In_2Se_3$. More interestingly, interfacial ferroelectricity could emerge by stacking non-polar materials in a symmetry-breaking way[24–32], e.g., 1T′-$WTe_2$, parallel-hexagonal boron nitride (hBN), R-type transition metal dichalcogenides (TMDs) and

[1]State Key Laboratory for Mesoscopic Physics, School of Physics, Peking University, Beijing, China. [2]Institute of Microelectronics, Chinese Academy of Sciences, Beijing, China. [3]University of Chinese Academy of Sciences, Beijing, China. [4]National Institute for Materials Science, Tsukuba, Japan. [5]School of Physics, Huazhong University of Science and Technology, Wuhan, Hubei, China. [6]School of Aerospace Engineering, Beijing Institute of Technology, Beijing, China. [7]School of Physical Sciences, University of Chinese Academy of Sciences, Beijing, China. [8]State Key Laboratory of Quantum Optics and Quantum Optics Devices, Institute of Opto-Electronics, Shanxi University, Taiyuan, China. [9]Collaborative Innovation Center of Extreme Optics, Shanxi University, Taiyuan, China. ✉e-mail: jmlu@pku.edu.cn

TMDs heterobilayers, greatly expanding the scope of ferroelectric candidates.

In particular, the most thoroughly studied non-polar carbon material – graphene – was also found to be ferroelectric:[33,34] a bilayer graphene aligned with hBN on both sides exhibits a (special) gate-specific ferroelectricity (GSFE) and concomitant anomalous screening ascribed to electron dynamics rather than purely ionic displacement. According to the interlayer charge transfer (ICT) model[33], the two-dimensional polarization $P_{2D}$ depends on the density of correlated electrons accommodated by an engineered moiré band, a flexibly controlled degree of freedom. However, the lack of moiré features in experimental samples so far has prevented quantitative comparison with the theoretical model, leaving open questions to the validity of the physical mechanism that is essential for extensive explorations of this novel family of interfacial ferroelectrics.

In this work, we take advantages of clear moiré peaks and Landau levels (LL) to quantitatively explore ferroelectricity and anomalous screening. Surprisingly, they could be observed even when the Fermi surface is deep into the dispersive bands that are beyond moiré bands, contradicting the prevailing ICT model where a half filling splits the flat moiré band and induces interlayer charge transfer. The polarized charge density – without the stringent constraint of moiré bands – was found to reach $10^{13}$ cm$^{-2}$. Opposite to previous observation, GSFE can be also switched off by the normal gate in a broad parameter space, while the anomalous screening keeps to be stable. Specifically, when the normal gate was set in a high-voltage range, the hysteresis loop obtained by scanning the special gate tends to collapse into the same states with anomalous screening. For device applications, GSFE is also

proved to act as a nonvolatile switch of intrinsic quantum properties such as topological valley current[35,36]. Our results have clarified important characteristics of GSFE, which are essential to unveil the physical mechanisms and promote potential applications of ferroelectric vdW devices with appreciable charge polarization.

## Results

### Untying the ferroelectricity and correlated electrons in moiré bands

The device D1 consists of a Bernal bilayer graphene sandwiched by two hBN flakes and graphite gates (Fig. 1a). Straight edges of graphene and hBN are used for crystallographic alignment. The resulting angles between graphene and top/bottom hBN, derived from optical and electrical characterizations (Supplementary Note 1), are respectively ~30° and ~0°, where the smaller angle enables the observation of clear moiré features acting as a marker to locate the Fermi surface during the ferroelectric hysteresis.

Figure 1b displays a normal transfer characteristic by sweeping the back gate $V_b$, i.e., the forward and backward curves coincide with each other. In contrast, the top gate $V_t$ is a special one that gives rise to ferroelectric hysteresis (Fig. 1c), where the resistive peaks at charge neutral point (CNP) and full filling point (FFP, see Supplementary Note 1.1 for the assignment) of a moiré band are delayed to appear. The delay is ~0.2 V nm$^{-1}$, slightly larger than the separation between CNP and FFP. Such a gate-specific hysteresis resembles that in ref.[33] and D2 in this study (Supplementary Note 2). Throughout the manuscript, we define $V_b$ as a normal gate and $V_t$ a special gate. To be complete, we measured the $V_t$-$V_b$ phase diagram by sweeping $V_t$ fast while increasing

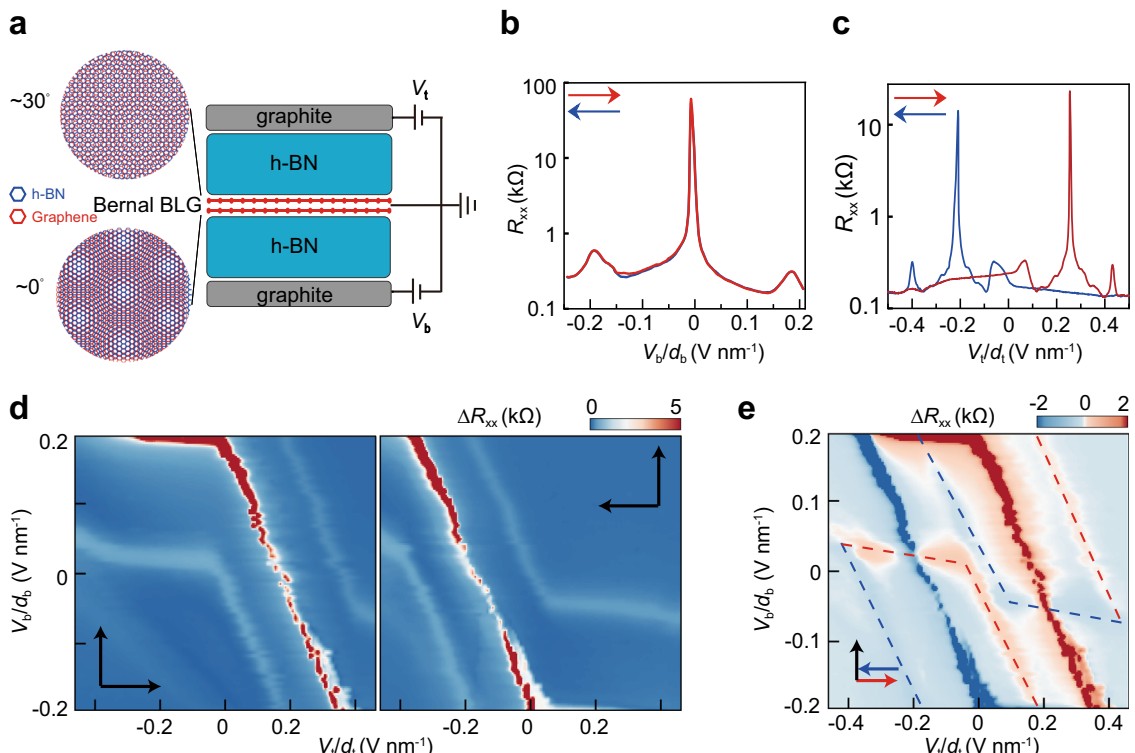

**Fig. 1 | Ferroelectricity in a bilayer graphene moiré superlattice. a** Schematic of a van der Waals heterostructure and the device measurement set-up, in which the graphene is roughly twisted by 0° and 30° with two hBN flakes, respectively. A moiré superlattice is thus formed at one interface and a close-quasicrystal at the other. $V_t$: the top gate voltage; $V_b$: the back gate voltage. **b** The transfer curves have no hysteresis between the forward and backward scanning of $V_b$. The second Dirac peak indicates the moiré superlattice is of 11.5 nm, corresponding to an angle of 0.69° between BLG and hBN. **c** A ferroelectric hysteresis is observed by sweeping $V_t$.

**d** A complete phase diagram of resistance is constructed by sweeping $V_t$ forth (left) and back (right) while increasing $V_b$ in a step-by-step fashion. Anomalous screening at both CNP and FFP could be identified by horizontal lines (red and white in color, respectively). **e** Difference in resistance between the two panels in **d**. Multiple hysteresis loops are formed by three pairs of blue and red peak lines at CNP and FFP. Note that the loops for electron and hole FFP (highlighted by dashed lines) are only half of a closed loop.

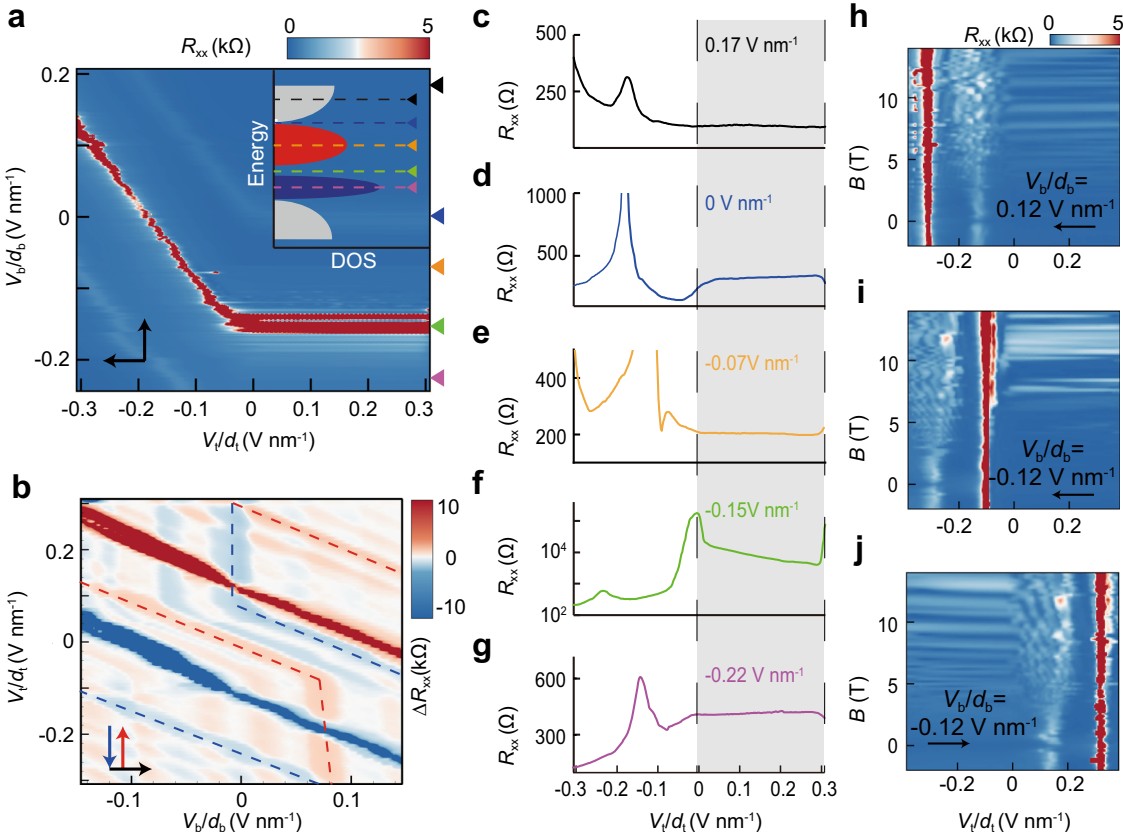

**Fig. 2 | Ferroelectricity and anomalous screening in magnetic fields. a** The anomalous screening at CNP and FFP is highlighted in a $V_t$-$V_b$ phase diagram, where five linecuts are denoted by colored triangles. Inset: Their correspondences in the band structure. **b** In perpendicular magnetic fields (10 T), anomalous screening behavior and the associated hysteresis loops become visible in many more electronic states, two of which are highlighted by the dashed curves. This is due to prominent Landau levels that enhance the visibility of low-resistance states. **c–g** Careful examination reveals that, a flat resistance curve independent on $V_t$ (*i.e.*,

anomalous screening) indeed exists in electronic states with Fermi surfaces higher than the electron moiré band **c**, at FFP **d**, within the electron moiré band **e**, at CNP **f**, and within the hole moiré band **g**. Importantly, these anomalous curves span over almost the same $V_t$ regimes highlighted by the grey shadow, despite their distinct displacement fields. **h–j** Landau fans are characterized by three typical electronic states, whose Fermi surfaces (when the anomalous screening occurs) are above **h** and within **i** the electron moiré band, and below the hole moiré band **j**, respectively.

$V_b$ step by step from −0.2 to 0.2 V nm⁻¹. Throughout the manuscript, the fast-scan gate is plotted in the $x$ axis and the slow one in the $y$ axis; the scanning directions are denoted by arrows in each graph. As shown in Fig. 1d, striking hysteresis (and anomalous screening manifested as horizontal ridges) are observed at both CNP (red) and two FFP (white). For better visualization, subtraction between the left and right panels is performed in Fig. 1e, where the central parallelogram composed of red and blue resistance peaks represents the hysteresis loop at CNP. There are another two replicas in light red and blue (highlighted by dashed lines) for two FFP, though only the lower (upper) half is formed on the electron (hole) side. It's surprising that the shift between the parallelogram and two replicas is basically along the vertical direction, *i.e.*, the turning into anomalous screening states depends merely on the special gate $V_t/d_t$ rather than a total displacement field $D=(D_b-D_t)/2$, where $D_{b(t)}=\varepsilon_0\varepsilon_r V_{b(t)}/d_{b(t)}$. More importantly, the hysteretic FFP on the hole side indicates that GSFE and anomalous screening exist even when the (possibly correlated) moiré band is empty. In the following we check their existence for dispersive bands at higher energy levels.

Figure 2a shows the phase diagram of a backward sweep, where five typical linecuts (colored triangles) correspond to representative Fermi surfaces (inset of Fig. 2a): higher than the electron moiré band (Fig. 2c), inside the electron moiré gap (Fig. 2d), within the electron moiré band (Fig. 2e), at CNP (Fig. 2f), or within the hole moiré band (Fig. 2g). All these states are driven out of the anomalous screening status (shaded region) by nearly the same $V_t/d_t$, consistent with the observation in Fig. 1e. To better present the behavior of low-resistance

states, perpendicular magnetic fields were applied to show the Landau levels. In Fig. 2h, the horizontal LL could be clearly identified above 5 T. Since the flat LL is ended close to CNP, the Fermi surface in the anomalous screening regime must stay within the electron moiré band. For Fermi surfaces beyond electron and hole moiré bands, LL fans are shown in Fig. 2i, j, respectively. Then, we fix **B** as 10 T and measure the $V_b$-$V_t$ phase diagram (Fig. 2b). Multiple hysteresis loops, both within and outside moiré bands, could be observed owing to the enhanced visibility by LL (See raw data in Fig. S10). To conclude, the access to the anomalous screening status does not rely on a specific Fermi surface within a moiré band.

Actually, there have been already some signatures for the absence of correlation, e.g., no correlated resistive peaks at half filling could be observed in the transfer curves shown in Fig. 1b. In another sample D2 with nearly identical properties (Supplementary Note 2), we even cannot see FFP peaks because of relatively larger twist angle, neither did previous reports of GSFE[33]. In addition, the moiré band may not be isolated from dispersive bands. As shown in Fig. S2a, b, we can only observe the sign reversal of carriers in LL fans of longitudinal resistance but not in Hall signals, which indicates the energy gap at FFP is not complete. The absence of a full gap also explains the relatively weak FFP peak in Fig. 1b.

## Strongly enhanced charge polarization
Before presenting experimental results, we first make a simple estimation of polarized charge carriers. Following the ICT model[33], a moiré

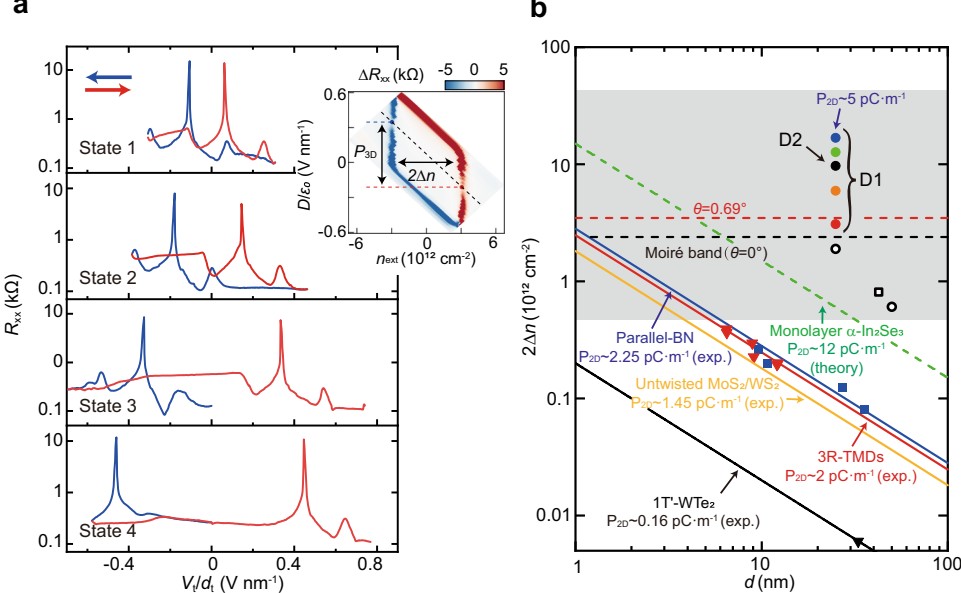

**Fig. 3 | Enhanced charge polarization. a** Four states with distinct magnitudes of hysteresis. Note the satellite peaks at FFP exhibit a constant shift from CNP in all states, indicating the moiré superlattice keeps unchanged. The sweeping curves of State 2 is also denoted as the black cutting line in the inset, where the polarized charge $\Delta n$ and the internal electric dipole $P_{3D}$ are also illustrated. **b** Summary of polarized charge densities in various ferroelectric systems. The $2\Delta n$ for D1 and D2 are depicted by filled circles, which are close to or well above the capability of typical moiré bands (the red dashed line: the twisted angle of 0.69° as in D1). GSFE

devices from literatures are plotted as open circles[33] and squares[34]. The horizontal grey background emphasizes that their polarization $\Delta n$ actually do not depend on the thickness of gating dielectrics $d$. However, for others integrated in the dielectric gate, the induced charge polarization in the conduction channel is inversely proportional to the total thickness of the gating dielectrics. 1T'-WTe$_2$, parallel-hBN and R-TMDs are taken from experiments, while monolayer $\alpha$-In$_2$Se$_3$ from theoretical calculations.

band would accommodate strongly correlated electrons[37,38], half of which transfer to another spatially separate graphene layer and eventually induce interlayer electric dipoles[33]. Take the twist angle of the D1 sample (0.69°) as an example, a half moiré band will accommodate electrons ~$1.7 \times 10^{12}$ cm$^{-2}$. Assuming the band polarization in the bilayer graphene is perfect, then the interlayer charge transfer $\Delta n$ is just equal to the half moiré band. Since the interlayer distance $d_{dipole}$ is 0.34 nm, the polarization would be $P_{2D} \sim e\Delta n \times d_{dipole} \sim 0.9$ pC m$^{-1}$. We note that in ref. 33. the difference in displacement fields between two gapless charge neutral points $P_{3D} = \varepsilon_0 \times \Delta(D/\varepsilon_0)$ was taken as an independent way to determine the remnant polarization $P_{2D} = P_{3D} \times d_{dipole} = \Delta D \times d_{dipole}$. However, according to the *standard* parallelogram depicted in the inset of Fig. 3a, one gets $\Delta D = e\Delta n$. Consequently, these two methods are not independent but rather exactly the same.

Experimentally, the charge polarization $2\Delta n$ in graphene can be conveniently measured from the shift of CNP between back and forth sweeping (the inset of Fig. 3a). During several thermal cycling of D1, we found the ferroelectric hysteresis may change its magnitude by several times (labeled as States 1–4 in Fig. 3a), but are qualitatively similar (See additional data for State 1 in Figs. S17, 18). Note that the moiré feature remains the same in all states. It's well known that thermal cycling may change the heterostructures via strain and twist angle[39,40], however, the twist angle of 0.69° responsible for the moiré superlattice seems not influenced. We then compare $2\Delta n$ for all the four states (filled circles) with the electron density accommodated by the moiré band (dashed horizontal lines) in Fig. 3b. Distinct from similar devices in previous reports where $\Delta n$ (open symbols) are only a small part of a moiré band, $\Delta n$ in State 1 (red circle) is in approximate to the half filling, and States 2–4 have much larger polarized charge density than the full filling (See Supplementary Note 1.1 for the identification of FFP). As for the most striking State 4, $\Delta n$ approaches $10^{13}$ cm$^{-2}$, which is translated as $P_{2D} \sim 5$ pC m$^{-1}$, significantly larger than the calculated 0.9 pC m$^{-1}$. The exceptionally high charge polarization questions the role of moiré bands in the ICT model[33].

Also plotted in Fig. 3b are ferroelectric van der Waals materials, including $\alpha$-In$_2$Se$_3$ that is arguably the strongest monolayer ferroelectric[8], and typical interfacial ferroelectricity[24,27,30] (1T'-WTe$_2$, parallel-hBN and R-TMDs). Note that here the $\Delta n$ for these systems are not intrinsically polarized charge within ferroelectrics but are sensed by a conduction layer, where $\Delta n = P_{2D}/ed$ with $d$ as the thickness of the total gating dielectric[27]. Were the $\Delta n$ observed in GSFE caused by parallel-stacking (*i.e.*, rhombohedral) of hBN layers, the large polarization would require each interface inside the hBN flake of the special gate to be such aligned, which is energetically unfavorable.

## Discussion about the physical mechanism

The origin of GSFE can be either electron dynamics (the ICT model in the bilayer graphene) or lattice distortion (sliding ferroelectricity within hBN flakes), both of which cannot explain all of the essential features observed in experiments. As discussed for the ICT model, the amount of transferred electrons can well exceed the capacity of a moiré band (Fig. 3) and the concomitant anomalous screening can occur outside the moiré bands, *i.e.*, within dispersive bands (Fig. 2), both suggesting that the mechanism does not rely on electronic correlation.

The sliding ferroelectricity, although excluded when confined within hBN flakes, was also proposed to take place at the interface between crystallographically aligned graphene and hBN[7]. Two polar states correspond to carbon atoms overlapping with either boron or nitrogen atoms; by expanding the area of one polar state (and shrinking the other correspondingly), the net polarization over a moiré supercell can be switched upwards or downwards (Figs. S20, S21 and associated discussions in Supplementary Information, section 4). The same order of $P_{2D}$ in GSFE (~5 pC m$^{-1}$) and prototypical sliding ferroelectrics (~2 pC m$^{-1}$) encourages further exploring the relation between the two. The drawback, however, lies in that (1) anomalous screening is more likely to be driven by electron dynamics; (2) the polar states are expected to survive only under perpendicular electric fields in ideal

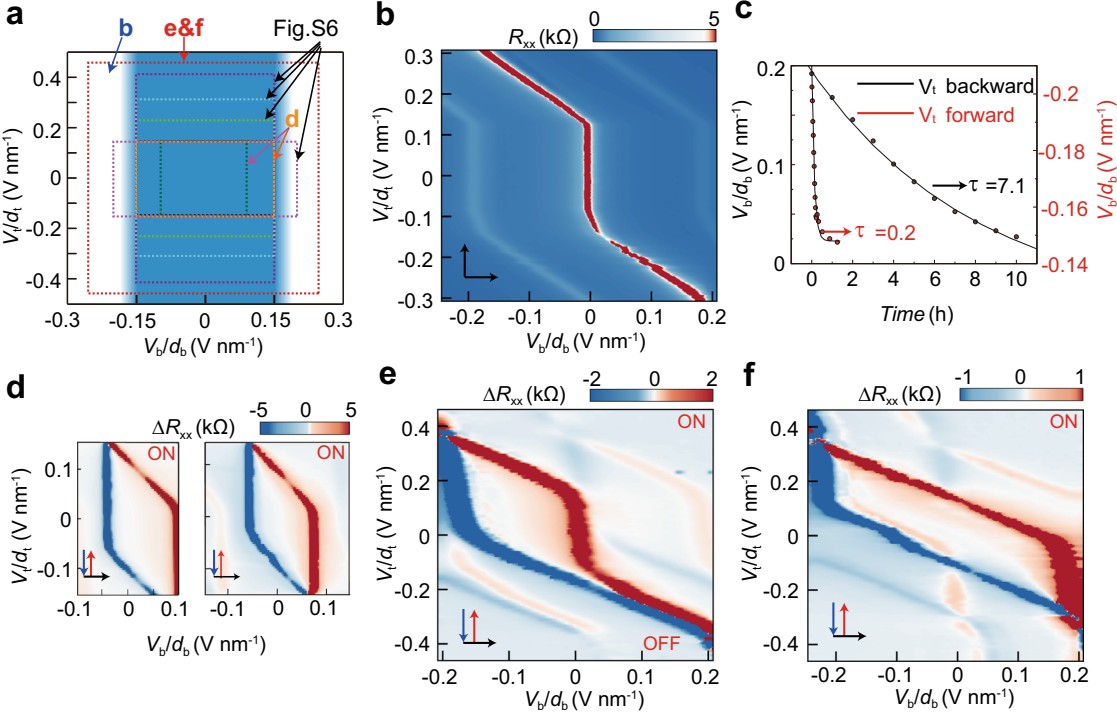

**Fig. 4 | Gate-specific ferroelectricity controlled by the other gate. a** Two regimes are identified according to the stability of the hysteretic behavior. Below the threshold of $V_b/d_b$ -0.15 V nm$^{-1}$ the ferroelectricity is stable (highlighted in blue), whereas it becomes more and more fragile when exceeding the threshold. The scanning ranges used in other panels and Fig. S6 are denoted. **b** Take the scanning range of $V_b/d_b$ as above 0.2 V nm$^{-1}$, one obtains a phase diagram without ferroelectric hysteresis by scanning $V_t/d_t$ slowly (see its hysteresis-free counterpart in Fig. S3). However, the anomalous screening remains robust. **c** The relaxation to the stable states shown in **b** is quantified by a temporal characterization. Two representative points are selected: one is for $V_t > 0$ in the backward scan and the other is $V_t < 0$ in the forward scan. The characteristic time fitted by an exponential decay function are 7.1 and 0.2 h, respectively. Measurement details and raw data can be found in Fig. S4. **d** Below the threshold, a regular parallelogram can be formed. An obvious criterion is that the resistance ridge at CNP in the anomalous screening regime is strictly parallel with the $y$ axis. Here the scanning range of $V_b/d_b$ is 0.1 (left) and 0.15 (right) V nm$^{-1}$. The label ON/OFF means hysteresis loops are switched on/off. **e** In a larger scanning range of $V_b$ -0.21 V nm$^{-1}$, the relaxation distorts the parallelogram in the upper-left corner of the $V_t$-$V_b$ phase diagram, *i.e.*, the ridge is no more in parallel with the $y$ axis; furthermore, it eliminates completely the hysteresis in the lower-right corner. **f** By increasing the scanning rate through a particular scheme (see details in Fig. S5), the relaxation could be overcome and hence a full hysteresis loop is restored.

cases; for practical samples with pinning centers the states may exist, but probably coexist during switching, contrasting a uniform ferroelectric polarization in GSFE (Fig. S1c).

Overall, to reconcile all the experimental findings, e.g., the giant charge polarization and anomalous screening, it may be necessary to take into account both the electron-driven ICT model and the lattice-driven sliding ferroelectricity. A convincing mechanism calls for more conclusive experimental features.

### The normal gate as a switch knob of the special gate

Figure 4a summarizes the scan ranges performed. In brief, within the regime of $V_b/d_b < 0.15$ V nm$^{-1}$ (highlighted in blue), a stable ferroelectricity could be observed. However, above this threshold value (determined in Fig. S7), the hysteresis loop is metastable: the forward and backward scan tend to collapse into the same stable states in Fig. 4b (see the vanishing hysteresis in Fig. S3). The above stability criterion seems not to depend on $V_t$ (Figs. S8, S9). To quantify the dynamic process, a relaxation time $\tau$ is derived by monitoring the evolution of CNP as a function of time that is fitted by an exponential decay function. As shown in Figs. 4c and S4, $\tau$ is around 7.1 and 0.2 hours for $V_t > 0$ and $V_t < 0$, respectively.

The strong dependence of relaxation time on the normal gate provides a convenient way to switch ferroelectricity. Firstly, when $V_b/d_b$ is relatively small, the resistance ridge of anomalous screening is strictly in parallel with the $V_t$ axis, forming a standard parallelogram (Fig. 4d with raw data in Fig. S6). Upon increasing the scanning range of $V_b$, e.g., 0.21 V nm$^{-1}$ in Fig. 4e, the relaxation to the

stable states in Fig. 4b becomes so significant that the CNP peak deviates from a vertical line gradually. The relaxation is much more serious for $V_t < 0$ ($\tau$-12 minutes), so the hysteresis almost disappears. To suppress the relaxation, in Fig. 4f the scanning rate is accelerated by a particular gate sweeping scheme (Fig. S5). As expected, a full hysteresis loop could be restored, resembling that in Fig. 1e.

### Ferroelectric switching of intrinsic quantum properties

We firstly examine the thermal stability. At various temperatures up to 300 K, typical phase diagrams are measured (Figs. S11–14), from each of which the magnitude of ferroelectric polarization could be extracted. As shown in Fig. 5a, the polarization remains stable below 100 K, but decays gradually at higher temperatures. The robustness in a wide range of temperature allows for investigating many quantum phenomena, e.g., the topological valley current. Following well-established measurement schemes[34,35], we obtain the local and nonlocal signals in Fig. 5b, c, respectively (See raw data in Fig. S15). To confirm the nonlocal results indeed originate from valley transport, we extract the resistance of CNP and compare their relationship with $R_{NL}$-$R_L^3$ in Fig. 5d. Apart from the low-resistance regime, the fitting is quite good over two decades, indicating that the bulk valley transport starts to dominate once the energy gap is opened at CNP. The results also suggest that the ferroelectricity is compatible with intrinsic properties of a bilayer graphene, opening more opportunities to manipulate these quantum phenomena with additional degree of freedom.

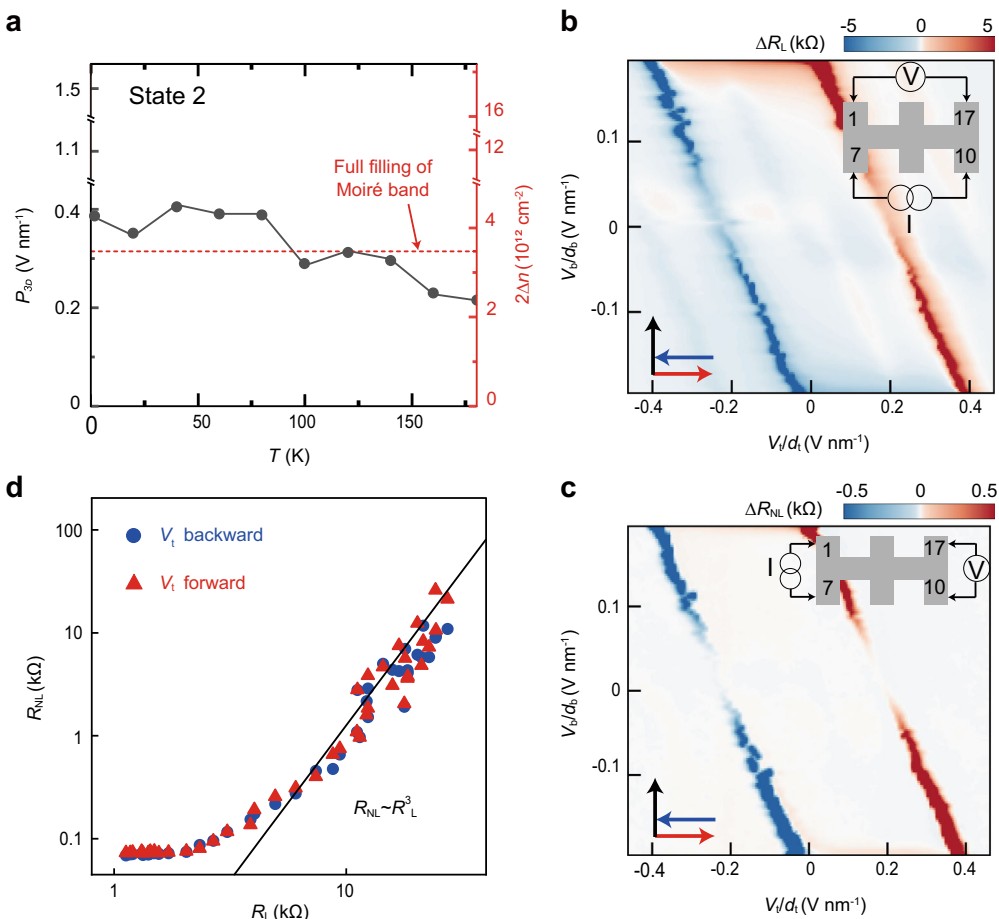

**Fig. 5 | Ferroelectric switching of topological valley current. a** The internal electrical polarization $P_{3D}$ (State 2) keeps almost the same below 100 K, but gradually decays at higher temperature. **b**, **c** Local **b** and nonlocal **c** signals are compared at 50 K. Only at CNP the nonlocal resistance could be observed. Inset: Measurement schemes. **d** The scaling of $R_{NL}$-$R_L^3$ is consistent with the topological valley transport.

## Discussion

To conclude, by using moiré features and Landau levels as indicators, we manage to prove that the gate-specific ferroelectricity does not rely on correlated electrons. Naturally, the charge polarization has been found to exceed the half filling of a moiré band. The maximum charge polarization achieved so far is approximately $10^{13} cm^{-2}$, which may be further improved by optimizing twist angles and constituent materials. For device applications, we find the normal gate actually acts as a switching knob of the functioning of the special gate; the wide compatibility allows for ferroelectric switching of various quantum phenomena including topological valley transport as demonstrated. Overall, our results disclose sophisticated functionalities of the gate-specific ferroelectricity and provide key clues of the underlying mechanism that may be essential to the discovery of a large family of ferroelectrics, promoting potential applications of van der Waals ferroelectric devices with a large charge polarization.

## Methods
### Sample fabrication

The bilayer graphene crystals, graphite gates, and boron nitride flakes were all mechanically exfoliated on to silicon wafers and identified with optical microscopy. The multilayer heterostructure was then fabricated following the standard dry transfer method[41]. Then reactive ion etching was used to pattern the Hall bar geometry. At last, one-dimensional edge contacts[42] were prepared following the standard e-beam lithography and e-beam evaporation of Cr (1 nm)/Au (50 nm). The device geometries are detailed in Fig. S1.

### Characterization of crystallographic direction

For graphene, samples with a pair of long and straight edges whose angles are 30° or 90° were selected and then 532 nm laser with polarization aligned in parallel with edges was used to obtain Raman spectra at D band (~1350 cm$^{-1}$). The intensity for zigzag edges is negligible, whereas that for an armchair edge is more significant[43]. For hBN flakes, the crystallographic axes were identified by Second Harmonic Generation (SHG)[44]. SHG signals were collected by WiTec UHTS300 with incident light wavelength of 1064 nm and a fixed excitation power of 20 mW. It is minimized when the polarization of incidence is along the zigzag and maximized for the armchair direction.

### Electrical measurement

All the transport measurements were carried out in a cryostat with base temperature of 1.5 K and a superconducting magnet up to 14 T. Unless specified otherwise, the sample temperature was at base temperature. A standard four-probe method of constant current was performed. The AC current was supplied by Stanford Research Systems SR830 lock-in amplifiers with a working frequency of 17.777 Hz. The DC gate voltages were output by two Keithley 2400 Source Meters (See device configurations in Fig. S1). The nonlocal measurement in Fig. 5 in the main text was carried out at 50 K following previous works[35,36].

### Reporting summary

Further information on experimental design is available in the Nature Research Reporting Summary linked to this paper.

## Data availability

All figures are provided in Source Data file. All other data that support the findings of this study are available from the corresponding author upon reasonable request. Source data are provided with this paper.

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

## Acknowledgements

J.L. acknowledges support from the National Key R&D Program of China (Grant nos. 2019YFA0307800 and 2021YFA1400100), Beijing Natural Science Foundation (grant no. Z190011), NSF of China (grant no. 11974027). C.H. acknowledges support from Beijing Natural Science Foundation (grant no. 4222084) and CAS Pioneer Hundred Talents Program.

## Author contributions

J.L. conceived the project. R.N. fabricated devices and performed transport measurements with assistance from Z.L., X.H., Z.Q., and Z.Y. Crystallographic characterization was performed by R.N., Z.L., Q.L., and T.L.; K.W. and T.T. synthesized boron nitride crystals. M.W., X.W., Q.R., and J.H. provided theoretical support. J.L., C.H., J.M., Z.H., K.L., Z.G. supervised the project. All authors contribute to the data analysis. R.N., and J.L. wrote the paper with input from all authors.

## Competing interests

The authors declare no competing interests.

## Additional information

**Correspondence and requests** for materials should be addressed to Jianming Lu.

