## [Peer Review File · Nature Communications]

Giant ferroelectric polarization in a bilayer graphene heterostructureREVIEWER COMMENTS

Reviewer #1 (Remarks to the Author):

From the measurements of Morie features and Landau Levels, Niu et al revealed the giant ferroelectric polarization in bilayer graphene which is encapsulated between BN layers. The polarization of ~ 5 pC/m is larger than most of the observed 2D ferroelectrics. It is also found that the ferroelectricity can be switched on or off via the normal gate, which is then used to modulate intrinsic properties such as topological valley current.

It is an interesting finding, however, I am not fully convinced by the work, mainly due to the unclear physics and origin of the giant ferroelectric polarization. I, therefore, recommend the major revision and re-assess afterward.

1. Figure 1a is very confusing. As indicated in the paper, the studied material system should be bilayer graphene sandwiched between BN layers, but it seems graphene+BN heterostructure sandwiched between h-BN layers in Fig. 1a, since the red hexagonal is graphene while the blue one is BN.

2. It will be absolutely essential to clearly explain why the forward and backward curves coincide with each other under the back gate V_b , but not under V_t as shown in Figs. 1b & 1c.

3. As discussed in the paper, the polarization of bilayer graphene should be around 0.9 pC/m based ICT model, while the net polarization from the sliding ferroelectricity should vanish when averaging over the Morie supercell, how does the giant polarization of 5 pC/m in this system? Detailed and convincing explanations are required.

Reviewer #2 (Remarks to the Author):

In the manuscript, Niu and co-workers present an impressive study of gate-specific ferroelectricity in precisely twisted graphene/BN heterostructures. With quantitative evaluation of the ferroelectric polarization and the moiré bands, several novel characteristics were disclosed, such as unexpectedly large polarization, independence on moiré bands and the additional control functionality of the so-called normal gate. The authors also provided a concrete example of ferroelectric switching of quantum properties. The work is essential to establish the physical mechanism of a new family of interfacial ferroelectrics and construct multifunctional quantum devices in the future.

Interfacial ferroelectricity by engineering van der Waals heterostructure is currently of intense interest, because of the promise to solve the famous thickness scaling problem in 3D ferroelectric films. Different from the sliding ferroelectricity, so far there has been no consensus on the mechanism of the gate-specific ferroelectricity. Thus, the topic of this work is highly relevant for the growing field of twisted van der Waals heterostructures. The results will be of interest to a broad readership in both 2D materials and ferroelectricity. Therefore, I recommend this paper for publication in Nature Communications.

Several questions for further improvement are listed as follows:

1. Layer-specific ferroelectricity can be generally categorized into two groups, one of which has a counterclockwise resistance loop and the other is clockwise. The authors mainly present the latter type of hysteresis in the main text due to the 0/30 degrees configuration of the studied devices. How about the 0/0 configuration? Would the authors comment on the relation between alignment angle configurations and the loop direction?

2. Besides the angle alignment, graphene ferroelectricity is also dependent on the Bernal bilayer graphene itself. An opposite case is the absence of ferroelectricity in monolayer graphene. As such, in the discussion part of physical mechanism, material dependence should be detailed.
 3. One of the main findings is that the ferroelectricity is not strictly connected to the moiré bands. Especially, in Fig. 3b the charge polarization obviously bypasses the full moiré band. Would the author comment the role of moiré superlattice?
 4. Another important tool to clarify the existence of anomalous screening beyond moiré bands is the Landau level fan. In previous literatures, LL was not identified. What's the reason of observing LL in this study?
 5. As shown by the comprehensive investigation in Table S1, the ferroelectricity is sensitive to alignment angle. AFM tip is a useful tool for in-situ operation of angle alignment, which would give more insight into relationship between twist angle and ferroelectricity. It is highly recommended in a future study. As for the present samples, e.g. in D1, there are many pairs of electrodes, it's reasonable to speculate that the angle may differ between different pairs. Is the ferroelectricity repeatable in other pairs?
 6. In line 45, authors mentioned out-of-plane ferroelectric "1T-MoTe2". While in the corresponding original paper (ref. 22 of this work), such materials are named as "distorted 1T-MoTe2", short for "d1T-MoTe2". Please keep the same notation as the original reference.
 7. In Fig. 2 (h)-(j), authors showed Landau fans for three different electronic states where the Fermi surfaces locates at different band. However, in Fig. 2(i) and (j), the value of V_b/d_b are same (both -0.12). Please check if there are any typos.
 8. Ferroelectricity in hetero-bilayer TMDs has been found, which is also cited in the introduction part. This highly relevant data is recommended to be summarized in Fig. 3b.
 9. In the inset of Fig. 3a, as to be consistent with $2*\delta_n$, δ_P should be replaced by $2*\Delta_P$.
- Overall, this is an important paper that should be published in Nature Communication after addressing the above comments.

Reviewer #3 (Remarks to the Author):

This paper presents transport measurements of hBN-encapsulated bernal-stacked bilayer graphene (BLG), in a double gating configuration. An important detail is that the top- and bottom hBN stacks are differently oriented with respect to the BLG, such that a long periodicity moiré pattern forms in the bottom interface due to a nearly zero stacking angle, whereas the relative angle at the top interface is around 30 degrees. Because of this, a long periodicity moiré pattern forms only at the bottom interface.

The authors report on the BLG longitudinal resistance (R_{xx}) for varying top- and bottom gates. Their data suggests that, within a certain regime, the bottom gate serves as a chemical potential knob with which to populate the moiré bands formed at the bottom BLG/hBN interface. On the other hand, the authors observe that the R_{xx} profile shifts towards negative or positive voltages, depending on how the top gate voltage is swept. This is interpreted as hysteresis associated to the emergence of ferroelectricity in the device.

By looking at the top-gate hysteresis loop for fixed bottom-gate voltage (chemical potential), the authors conclude that this ferromagnetism is observed for any filling factor of the electron- and hole moiré bands, indicating that the effect is not related to a particular

correlated state emerging in these bands, but to a robust ferroelectric charge transfer across the structure. This charge transfer is estimated by the authors to be much larger than what has been either measured or observed in other ferroelectric 2D materials, as well as in correlated states in previously studied aligned hBN/BLG/hBN structures. This, together with their observation that the effect can be effectively switched on/off by setting the bottom gate voltage inside/outside a measured range of values makes this device a candidate for technological applications.

Overall, I find this work quite interesting, and believe it will be of interest to the broader 2D materials community. The authors' reported experiments are, in my opinion, convincing. The only shortcoming of this paper is that no attempt has been made to explain the origin of the ferroelectric effect in this system, which I believe somewhat diminishes its message. Can the authors comment on this?

In any case, I believe that what the authors have presented is novel, interesting, and experimentally rigorous enough to merit publication in Nat. Commun.

We thank the Referees for their critical reading and constructive comments. Before the point-by-point reply, we'd like to address the issue about physical mechanisms.

Generally speaking, ferroelectricity could originate from pure electron dynamics or spontaneous lattice distortion. For the device studied in this work, there are crystallographically aligned bilayer graphene and hBN flakes, thus we discuss all possible electron- and/or lattice-driven mechanisms in terms of two constituent layers and the interface.

a) Correlation induced interlayer charge transfer (within the bilayer graphene)

In this scenario, the essential object is the bilayer graphene, while the hBN flakes merely provide a special electrostatic environment. The bilayer graphene, subjected to perpendicular electric fields and a periodic superlattice potential, is endowed with layer-polarized moiré bands, *i.e.*, the electron and hole mini-bands reside in the bottom and top graphene, respectively. Electrons may exhibit strong correlation due to the narrow bandwidth of moiré bands. As shown in Fig. R1a, electrons firstly fill the valence band (the top layer in real space). Upon the band is half filled, the on-site energy U excludes double occupation within each moiré supercell, which is translated as band splitting in the momentum space. Compared with the energy gap Δ between conduction and valence bands, a larger U will result in prior population of the conduction band (the bottom layer in real space). In other words, electrons transfer from the top to the bottom layer, generating electric dipoles ordering between the two layers.

The above interlayer charge transfer model is able to qualitatively explain several key experimental observations, such as the difference between normal and special gates, the accompanied anomalous screening and so on. For the different gates, there may be only one hBN flake that forms a strong moiré pattern with graphene, whose corresponding mini-band is narrow enough to introduce correlation for further Hubbard band splitting. Consequently, this gate behaves as a special gate and the other as a normal one. For the anomalous screening, during the process of band splitting, the gate voltage (of the special gate) needs to compensate the rising chemical potential, so that no additional carriers are introduced into the system, mimicking the 'stop-working' behavior. As a result, this scenario was previously believed to account for the moiré ferroelectricity, despite the lack of moiré features in experiments.

However, as pointed out in the manuscript, the basis of the above ICT model – correlated electrons – has not been scrutinized yet. Our devices with clear moiré features severely contradict with the moiré physics on a quantitative level. Firstly, the amount of transferred electrons is found to depend on specific status of the device (Fig. 3), which can far exceed the capacity of a moiré band. Such a quantitative comparison was in lack in literatures due to the missing moiré features. Secondly, the concomitant anomalous screening can occur outside the moiré bands (on both electron and hole sides), *i.e.*, within dispersive bands (Fig. 2), which is against electronic correlation.

The new findings are in discrepancy with moiré bands of correlated electrons, challenging the ICT model. Nevertheless, an electron-driven mechanism remains possible if some new mechanisms can induce substantial charge transfer within a bilayer graphene.

Figure R1 Illustration of two types of ferroelectricity. **a.** Interlayer charge transfer model. The isolated moiré bands in the conduction (red) and valence (blue) bands are localized in the bottom and top layer in real space, respectively. The high-energy dispersive bands are omitted for clarity. Here the band gap Δ at charge neutral point is due to the perpendicular electric field, the correlation gap (on-site energy) U originates from the narrow bandwidth of a moiré band. The band splitting may induce electron transfer from one layer to another, forming spatially ordered electric dipoles. **b.** Sliding ferroelectricity at the interface of graphene and hBN flakes. When the twist angle is small or even zero degrees, relatively large domains (\sim several nanometers) of polar states can be formed. Around the red (blue) circle where C atoms locate just above N (B) atoms, the π electron cloud of C atoms is repelled (elongated) to form a polar state. In contrast, in the yellow circle such polarization cancels out over a unit cell of graphene. With external electric fields along a perpendicular direction, the domain areas may be enlarged or squeezed, leading to a net polarization over a moiré supercell.

b) Sliding ferroelectricity (at the interface between graphene and hBN)

It is also possible that the ferroelectricity stems from lattice distortion, just like almost all conventional ferroelectric materials. Following the well-established sliding ferroelectricity, we plot graphene and the crystallographically aligned hBN in Fig. R1b. Note that only the adjacent layer of graphene to hBN is considered, whereas the other layer of graphene is omitted because it is much less influenced by the hBN owing to a large distance.

Inside a moiré supercell, there are two polar states at the graphene/hBN interface (Ref. 7): when half of the C atoms of the graphene layer locate right over the N atoms of hBN layer, the π electron cloud of the C atoms will be repelled by the N atoms with negative charge, giving rise to a vertical polarization downwards; in contrast, when half of the C atoms of the graphene layer locate right over the B atoms, the π electron cloud of the C atoms will be attracted and prolonged by the B atoms with positive charge, giving rise to a vertical polarization upwards. Note that when both B and N are overlaid by C atoms, the polarization is averaged out over the unit cell of graphene.

Although the two polar states could cancel each other over a moiré supercell, untwisted graphene/hBN interface may also exhibit ferroelectricity similar to the twisted bilayer hBN (Ref. 27-29), where the macroscopic polarization is switched by expanding/shrinking the area of polar states through domain wall motions. Nevertheless, this lattice driven mechanism is difficult to explain the accompanied anomalous screening, as the latter is a kind of electron dynamics. There is another indirect evidence. Since the switching between two polar states is via domain wall motion, the two polar states of practical samples may coexist. The coexistence can be directly observed by microscopic characterization or – more easily – by electrical measurements. For example, in twisted bilayer hBN (Ref. 27) double resistance peaks always appear in the phase diagram, each of which represents either an upwards or downwards polar state. However, in our own samples and similar ones in the literature, such coexisting signals have never been observed. Instead, the polar states are verified to be quite uniform over the entire sample (Fig. R2 in the point-to-point reply). All of these observations, although indirect, indicate that the graphene ferroelectricity is unlikely to be from a pure lattice distortion.

c) Sliding ferroelectricity (within hBN flakes)

The hBN flake may have stacking faults, invoking the sliding ferroelectricity naturally. However, this mechanism can be excluded completely. If such an imperfect hBN exists, both top and bottom hBN cut from the same thin crystal would enable hysteresis of resistivity, in direct contrast to the experimental results. Quantitatively, the polarized carrier in graphene is too large to be from sliding ferroelectricity in hBN because, as estimated in the paragraph 3, page 6 of the revised manuscript, each layer of the hBN flake needs to be rhombohedrally stacked.

To summarize, our results demonstrate that neither the electron-driven ICT model nor the lattice-driven sliding ferroelectricity can solely explain all the observation, hence the combination of the two may be required for a complete model.

In the following, we address all the questions point by point.

Reviewer #1 (Remarks to the Author):

From the measurements of Morie features and Landau Levels, Niu et al revealed the giant ferroelectric polarization in bilayer graphene which is encapsulated between BN layers. The polarization of ~ 5 pC/m is larger than most of the observed 2D ferroelectrics. It is also found that the ferroelectricity can be switched on or off via the normal gate, which is then used to modulate intrinsic properties such as topological valley current.

It is an interesting finding, however, I am not fully convinced by the work, mainly due to the unclear physics and origin of the giant ferroelectric polarization. I, therefore, recommend the major revision and re-assess afterward.

Reply: We thank the referee for the constructive suggestion. A thorough discussion about the experimental observation and theoretical models is now presented in the first part of this reply.

In response, we add a section in page 6 in the revised manuscript (highlighted in red) and more details in Supplementary Information, section 4.

1. Figure 1a is very confusing. As indicated in the paper, the studied material system should be bilayer graphene sandwiched between BN layers, but it seems graphene+BN heterostructure sandwiched between h-BN layers in Fig. 1a, since the red hexagonal is graphene while the blue one is BN.

Reply: We thank the referee for pointing out the misleading color in Fig. 1a. To highlight moiré patterns at the interface, red and blue colors were used to denote graphene and hBN, respectively. Now in the revised manuscript, both layers of the bilayer graphene are plotted in red.

2. It will be absolutely essential to clearly explain why the forward and backward curves coincide with each other under the back gate V_b , but not under V_t as shown in Figs. 1b & 1c.

Reply: The gate-specific behavior is indeed one of the most significant features. One is the special gate that exhibits ferroelectric hysteresis (V_t in Figs. 1c), the other is a normal one without hysteresis (V_b in Figs. 1b). At first glance, the two gates should be equivalent. Actually, the details – twist angle and interaction strength – of the interface between graphene and hBN for the two gates may be not the same.

Corresponding to the two possible scenarios (in the first part of this reply), the requirement of a special gate is as follows: (1) In the ICT model, a possible explanation of distinct functionality is that the top and bottom hBN does not provide equally strong periodic modulation of electrostatic potential. The stronger one that controls the Hubbard band splitting, anomalous screening and ferroelectricity, would act as a special gate. (2) In the scenario of sliding ferroelectricity, the special gate must be associated with a polarized interface between hBN and graphene, *e.g.*, two polar states may arise due to the inequivalent influence of B and N on the π electron cloud of C atoms.

Despite that a unified theory beyond the above two scenario remains elusive, it is believed that the electronic structure of the interface should be in charge of the performance of special and normal gates.

In response, we put the above discussion in section 4 of Supplementary Information (paragraph 3, page 7).

3. As discussed in the paper, the polarization of bilayer graphene should be around 0.9 pC/m based ICT model, while the net polarization from the sliding ferroelectricity should vanish when averaging over the Morie supercell, how does the giant polarization of 5 pC/m in this system? Detailed and convincing explanations are required.

Reply: Before comparing experimental observation and theoretical models, we apologize for

the mistake made in the previous version of manuscript, where we wrongly claimed that the net polarization *always* vanishes over the moiré supercells. Actually, a perpendicular electric field can expand/shrink the domain size of polar states, so that the net polarization survives and could be switched. Now it has been corrected in paragraph 2, page 7 of the revised manuscript.

To find the origin of experimentally observed giant polarization, we need to examine the two possible scenarios more carefully. However, our conclusion remain the same: both ICT and sliding ferroelectricity cannot quantitatively explain our experimental results. As also pointed out by the referee, there is a large difference in polarization magnitudes. It can be taken as a quantitative evidence for the invalidity of the prevailing ICT model. The giant charge polarization (Fig. 3), together with the observation of anomalous screening beyond the moiré band (Fig. 2), suggests that correlated electrons in the moiré bands cannot account for the graphene ferroelectricity, which is one of the main findings of this work. The magnitude, however, is relatively closer to that of the sliding ferroelectricity (~ 2 pC/m). Hence we need to discuss the possibility of sliding ferroelectricity at the interface between untwist graphene and hBN flakes. Nevertheless, as shown in the first part of the reply, it has its own weak point: a lattice-driven mechanism alone cannot explain anomalous screening, another significant feature of gate-specific ferroelectricity. Consequently, we suggest that a plausible scenario should involve both lattice- and electron-driven mechanisms, which may give convincing explanation of the observed charge polarization. The establishment of such a unified theory needs more theoretical and experimental work in the future.

Reviewer #2 (Remarks to the Author):

In the manuscript, Niu and co-workers present an impressive study of gate-specific ferroelectricity in precisely twisted graphene/BN heterostructures. With quantitative evaluation of the ferroelectric polarization and the moiré bands, several novel characteristics were disclosed, such as unexpectedly large polarization, independence on moiré bands and the additional control functionality of the so-called normal gate. The authors also provided a concrete example of ferroelectric switching of quantum properties. The work is essential to establish the physical mechanism of a new family of interfacial ferroelectrics and construct multifunctional quantum devices in the future.

Interfacial ferroelectricity by engineering van der Waals heterostructure is currently of intense interest, because of the promise to solve the famous thickness scaling problem in 3D ferroelectric films. Different from the sliding ferroelectricity, so far there has been no consensus on the mechanism of the gate-specific ferroelectricity. Thus, the topic of this work is highly relevant for the growing field of twisted van der Waals heterostructures. The results will be of interest to a broad readership in both 2D materials and ferroelectricity. Therefore, I recommend this paper for publication in Nature Communications.

Reply: We thank the referee for positive comments and recommendations.

Several questions for further improvement are listed as follows:

1. Layer-specific ferroelectricity can be generally categorized into two groups, one of which has a counterclockwise resistance loop and the other is clockwise. The authors mainly present

the latter type of hysteresis in the main text due to the 0/30 degrees configuration of the studied devices. How about the 0/0 configuration? Would the authors comment on the relation between alignment angle configurations and the loop direction?

Reply: We thank the referee for careful reading. In addition to the device of 0/30 configuration (D1, D2 in this study) whose performance (anticlockwise) is highly consistent with the previous reports, we also made 0/0 devices as shown in Supplementary Table S1. Unfortunately, we cannot find any signatures of ferroelectricity. Nevertheless, we believe both clockwise and anticlockwise loops could find their origin in the device structure rather than intrinsic properties of ferroelectrics (see below).

A ferroelectric field-effect transistor is conventionally made of a ferroelectric insulator as a gate and a semiconductor as a channel (Fe-FET); it can also be integrated by a ferroelectric semiconductor as a channel and an insulator as a gate (FeS-FET). Here in graphene ferroelectric devices, the hBN is the normal insulator and the bilayer graphene is the ferroelectric semiconductor, so it belongs to FeS-FET.

As shown in Ref. 15 (*Nat. Electron.* **2**, 580 (2019)), different working mechanisms result in distinct performances. While Fe-FET always exhibits anticlockwise loops, FeS-FET can show both chiral directions depending on a specific structural parameters, *i.e.*, the effective gate thickness. Such an explanation holds for all devices with graphene ferroelectricity: a thin hBN and an anticlockwise loop for D1 and D2 in this study, H2, H5 and H6 in Ref. 33; a thick hBN/SiO₂ and a clockwise loop for H4 in Ref. 33 and the device in Ref. 34. Interestingly, such a high consistency in turn supports our argument that the ferroelectric polarization occurs in the bilayer graphene and/or its surface rather within hBN flakes.

2. Besides the angle alignment, graphene ferroelectricity is also dependent on the Bernal bilayer graphene itself. An opposite case is the absence of ferroelectricity in monolayer graphene. As such, in the discussion part of physical mechanism, material dependence should be detailed.

Reply: We thank the referee for constructive suggestions. Indeed, besides Bernal bilayer graphene, twisted bilayer graphene (*APS March Meeting S60.010 (2022)*) and hBN intercalated bilayer graphene (*ArXiv210212398 Cond-Mat (2021)*) also exhibit very similar ferroelectricity. The diversity of these materials is very important to identify whether correlation is the key ingredient: While the former definitely has correlated electrons, the latter is expected to be free from electronic correlation. Therefore, graphene ferroelectricity seems to be independence of correlation, in line with our experimental results.

In response, in the revised manuscript we append the material dependence to support our conclusion in the discussion about the physical mechanism in section 4 of Supplementary Information (paragraph 2, page 6).

3. One of the main findings is that the ferroelectricity is not strictly connected to the moiré

bands. Especially, in Fig. 3b the charge polarization obviously bypasses the full moiré band. Would the author comment the role of moiré superlattice?

Reply: We agree with the referee that, the moiré band created by periodic superlattices could not provide sufficient electrons to account for the giant charge polarization in experiments. However, a small-angle twisted interface and thus the resultant moiré superlattice are still essential to observe ferroelectricity in a graphene heterostructure. The opposite cases are various control devices (without ferroelectricity) from DG8 to DG10 shown in the Supplementary Table S1, whose twist angles are obviously much larger than 5 degrees. In fact, we also made tens of devices with arbitrary twist angles (not shown in the table), none of them exhibit ferroelectricity.

It's important to stress that a device without moiré features does not mean the absence of moiré superlattices: the twisted angle that could be detected by electric measurement for our devices is less than ~2 degrees, due to the typical breakdown voltage of hBN flakes. For instances, in D2 (this study), the twist angle is around 2.7 degrees determined by optical measurement, but moiré features are in lack in resistance measurements. In such a device, a moiré superlattice is still present, although with a shorter superlattice constant and probably without electronic correlation.

In response, we add the above discussion to section 3 of Supplementary Information.

4. Another important tool to clarify the existence of anomalous screening beyond moiré bands is the Landau level fan. In previous literatures, LL was not identified. What's the reason of observing LL in this study?

Reply: We thank the referee for pointing out the importance of LL in ferroelectric devices. Indeed, LL was absent in previous reports, even in magnetic fields as high as 10 T. The absence is not limited by electron mobility, but may be ascribed to the bad contact due to the energy gap of a Bernal bilayer graphene opened by perpendicular displacement field. A good example could be found in *Science* **345**, 61 (2014), where a global back gate was used to lower the contact resistance of invasive fingers and improve the quantum Hall measurement significantly.

5. As shown by the comprehensive investigation in Table S1, the ferroelectricity is sensitive to alignment angle. AFM tip is a useful tool for in-situ operation of angle alignment, which would give more insight into relationship between twist angle and ferroelectricity. It is highly recommended in a future study. As for the present samples, e.g. in D1, there are many pairs of electrodes, it's reasonable to speculate that the angle may differ between different pairs. Is the ferroelectricity repeatable in other pairs?

Reply: We thank the referee for constructive suggestions for highly controlled experiments in the future. Indeed, there are many pairs of electrodes in D1, whose performances are almost the same with that shown in the main text. To be complete, we summarize several transfer curves in Fig. R2, which is also added as Fig. S1c in the revised manuscript.

Figure R2 Uniform ferroelectric hysteresis over the entire sample (D1). Three pairs of voltage probes show almost the same CNP and moiré features, although on the electron side there is a little shift.

6. In line 45, authors mentioned out-of-plane ferroelectric “1T-MoTe2”. While in the corresponding original paper (ref. 22 of this work), such materials are named as “distorted 1T-MoTe2”, short for “d1T-MoTe2”. Please keep the same notation as the original reference.

Reply: We thank the referee for pointing out the typos. In the revised manuscript, the notation has been corrected.

7. In Fig. 2 (h)-(j), authors showed Landau fans for three different electronic states where the Fermi surfaces locates at different band. However, in Fig. 2(i) and (j), the value of V_b/d_b are same (both -0.12). Please check if there are any typos.

Reply: We thank the referee for careful reading. We verify that Fig. 2i and j share the same value of V_b/d_b . The difference is due to the hysteresis between forward (Fig. 2j) and backward (Fig. 2i) sweeping.

8. Ferroelectricity in hetero-bilayer TMDs has been found, which is also cited in the introduction part. This highly relevant data is recommended to be summarized in Fig. 3b.

Reply: We thank the referee for the improvement of Fig. 3. In the revised version, corresponding data for the hetero-bilayer has been added (see the yellow line in Fig. R3b).

Figure R3 New data for untwisted MoS₂/WS₂ (yellow) has been added in the right panel.

9. In the inset of Fig. 3a, as to be consistent with $2 \cdot \Delta n$, ΔP should be replaced by $2 \cdot \Delta P$.

Reply: We thank the referee for critical reading. As the factor 2 can be absorbed into the distance between positive and negative charges, the present form will be kept to be consistent with previous reports. Nevertheless, we change it to P_{3D} to emphasize its difference with frequently used P_{2D} .

Overall, this is an important paper that should be published in Nature Communication after addressing the above comments.

Reply: We thank the referee for recommendation.

Reviewer #3 (Remarks to the Author):

This paper presents transport measurements of hBN-encapsulated bernal-stacked bilayer graphene (BLG), in a double gating configuration. An important detail is that the top- and bottom hBN stacks are differently oriented with respect to the BLG, such that a long periodicity moiré pattern forms in the bottom interface due to a nearly zero stacking angle, whereas the relative angle at the top interface is around 30 degrees. Because of this, a long periodicity moiré pattern forms only at the bottom interface.

The authors report on the BLG longitudinal resistance (R_{xx}) for varying top- and bottom gates. Their data suggests that, within a certain regime, the bottom gate serves as a chemical potential knob with which to populate the moiré bands formed at the bottom BLG/hBN interface. On the other hand, the authors observe that the R_{xx} profile shifts towards negative or positive voltages, depending on how the top gate voltage is swept. This is interpreted as hysteresis associated to the emergence of ferroelectricity in the device.

By looking at the top-gate hysteresis loop for fixed bottom-gate voltage (chemical potential), the authors conclude that this ferromagnetism is observed for any filling factor of the electron- and hole moiré bands, indicating that the effect is not related to a particular correlated state emerging in these bands, but to a robust ferroelectric charge transfer across the structure. This charge transfer is estimated by the authors to be much larger than what has been either measured or observed in other ferroelectric 2D materials, as well as in correlated states in previously studied aligned hBN/BLG/hBN structures. This, together with their observation that the effect can be effectively switched on/off by setting the bottom gate voltage inside/outside a measured range of values makes this device a candidate for technological applications.

Overall, I find this work quite interesting, and believe it will be of interest to the broader 2D materials community. The authors' reported experiments are, in my opinion, convincing. The only shortcoming of this paper is that no attempt has been made to explain the origin of the ferroelectric effect in this system, which I believe somewhat diminishes its message. Can the authors comment on this?

Reply: We thank the referee for positive comments and constructive suggestions. We agree with the referee that it is important to have a detailed discussion involving the present experiments and theories. As shown in the first part of this reply, so far there have been several possible scenarios, but a unified theory accounting for all the important features remains elusive. The quantitative characterization in this work is expected to inspire such a theory and/or to narrow down the list of various theoretical proposals in the future. Here we'd like to clarify several important questions.

a) Does electronic correlation play an essential role?

Electronic correlation is the driving force of interlayer charge transfer within a bilayer graphene. While the ICT model can qualitatively explain the essential features associated with the ferroelectricity such as specific-gate dependence and anomalous screening, its validity has not quantitatively checked. In devices with clear moiré features, we find that (1) the polarized charge density is too large to be accommodated by a moiré band; (2) the anomalous screening survives even when the Fermi level is beyond the moiré feature (*i.e.*, deep into the dispersive band). Actually, the phenomena seems to be independent on whether the electron is in the moiré or dispersive bands. The two facts cast doubts on the foundation of the present ICT model. A plausible theory should invoke another mechanism rather than electronic correlation to induce the interlayer charge transfer.

b) Is it due to a lattice-driven mechanism?

As the Bernal-type AB bilayer graphene is the most stable crystal structure, the lattice distortion can only occur at the interface of graphene/hBN or within hBN flakes. Similar to conventional ferroelectrics, polarization switching is mediated by domain wall motion. As the domain wall could be pinned by defects in practical samples, two polar states usually coexist during the switching process, which was indeed verified in sliding ferroelectricity (twisted bilayer hBN in Ref. 27). The absence of coexisting states and uniform ferroelectricity over the entire sample (Fig. R2) suggest that graphene ferroelectricity is unlikely to be from a lattice distortion.

Another difficulty is that, as mentioned previously, anomalous screening is more possible to stem from electronic dynamics.

c) Is a moiré superlattice important?

In theory, a long-period moiré superlattice is essential in the above two types of mechanisms: it either creates moiré bands in the ICT model or locally polarized states within a moiré supercell. From experimental points of view, a small twist angle is necessary but not sufficient for observing ferroelectricity, although the twist angle can be larger than the electrical detection range ~ 2 degrees (due to the limited breakdown voltage of hBN). The exact range of twist angles for observing ferroelectricity remains to be determined.

In response, a section summarizing the above questions has been added in page 6-7 in the revised manuscript (highlighted in red). A full discussion can be found in Supplementary Information, section 4.

In any case, I believe that what the authors have presented is novel, interesting, and experimentally rigorous enough to merit publication in Nat. Commun.

Reply: We thank the referee for positive comments and recommendation.

REVIEWER COMMENTS

Reviewer #1 (Remarks to the Author):

Although the authors provided additional discussions and explanations for the origins of the giant ferroelectricity in bilayer graphene/BN sandwiched structure, the combination effect of the ICT model and sliding ferroelectricity has been proposed and added to the revised manuscript, it is still not convincing enough in my opinion, for the reason as below:

The sliding ferroelectricity indicated in Fig. R1b implies that the polarization in the blue area should be upwards while that in the red area is downwards. They should be canceled out among the whole Morie lattice, and have zero contribution to the overall polarization. As the authors also pointed out, the signals of the two polarization states' coexistence were not observed.

It is argued that the domain areas may be enlarged or squeezed under the electric field along the perpendicular direction, leading to a net polarization. However, the evidence is missing.

I would suggest the authors provide evidence of the enlarged/squeezed domain under the electric field to rationalize the findings and estimate the polarization strength/values of the red/blue regions of Fig. R1b from the first principles simulations, namely where C atoms locate just above N (B) atoms.

A clear understanding of the mechanism is essential to improve the quality of the present work, and refresh the understanding for the intriguing phenomena.

Reviewer #2 (Remarks to the Author):

I read through the revised manuscript and the response letter. The authors fully addressed my comments. I thus recommend a publication.

Reviewer #3 (Remarks to the Author):

In my first report I recommended this paper for publication in Nat. Comms., only requesting that the authors commented on the microscopic origin of the ferroelectricity that they measure in the graphene-hBN heterostructures described in their manuscript. As a response, the authors have supplemented their manuscript with a detailed discussion of possible mechanisms behind this ferroelectric effect. I'm satisfied with this and my recommendation to publish the paper stands.

We thank the Referees for their constructive comments and recommendations. In the following, we address all the questions point by point.

Reviewer #1 (Remarks to the Author):

Although the authors provided additional discussions and explanations for the origins of the giant ferroelectricity in bilayer graphene/BN sandwiched structure, the combination effect of the ICT model and sliding ferroelectricity has been proposed and added to the revised manuscript, it is still not convincing enough in my opinion, for the reason as below:

The sliding ferroelectricity indicated in Fig. R1b implies that the polarization in the blue area should be upwards while that in the red area is downwards. They should be canceled out among the whole Morie lattice, and have zero contribution to the overall polarization. As the authors also pointed out, the signals of the two polarization states' coexistence were not observed.

It is argued that the domain areas may be enlarged or squeezed under the electric field along the perpendicular direction, leading to a net polarization. However, the evidence is missing.

I would suggest the authors provide evidence of the enlarged/squeezed domain under the electric field to rationalize the findings and estimate the polarization strength/values of the red/blue regions of Fig. R1b from the first principles simulations, namely where C atoms locate just above N (B) atoms.

Reply: We thank the referee for constructive comments. Following the referee's suggestion, we carried out DFT calculations to evaluate the energy difference between the two polarized states (*i.e.* C atoms on top of B and N, respectively), and find that a perpendicular electric field can change the sign of the energy difference, which theoretically rationalizes the nonzero net polarization in a moiré superlattice. In addition, the polarization difference between the two states is estimated to be in the order of 1.83 pC/m, similar to that of twisted MoS₂.

The calculation details and results are as follows.

- **Structure**

Two different stacking configurations of graphene and hBN are considered. The AB-stacking means the C atoms directly stacks on the B atoms and the N atoms stacks on the center of the hexagon in graphene. Similarly, the BA-stacking means the C atoms directly stacks on the N atoms and the B atoms stacks on the center of the hexagon in graphene. The crystal structures of AB-stacking and BA-stacking of a graphene/hBN vdW heterostructure are shown in Figure R1.

Figure R1. Atomic structures of AB stacking in a graphene/hBN heterostructure with top view (a) and side view (b), and BA-stacking with top view (c) and side view (d).

● Results

We calculated the energy difference ΔU between AB-stacking and BA-stacking in the graphene/hBN vdW heterostructure under different electric fields. As shown in Figure R2, it's obvious that the energy of AB-stacking and BA-stacking is different under the same electric field. When the electric field is small, the energy of AB stacking is always lower than that of BA stacking. So the area of AB stacking (blue color in the right insets) is greatly enlarged, and the BA stacking (red color in the right insets) shrinks simultaneously. When the upward electric field continues to increase, ΔU changes from a negative value to a positive value, which means the energy of AB stacking becomes higher than that of BA stacking. Hence the proportion of AB and BA stacking areas is reversed.

Similarly, the electrically tunable lattice relaxation and resulting polarization for MoS₂ bilayer have been simulated in a recent work (*npj 2D Mater. Appl.* **6**, 7 (2022)), where the domains with opposite polarizations relax unevenly (one enlarged and the other one squeezed) under external electric fields, giving rise to a non-zero net polarization (Fig. R3a-d). The response of such polarization to the electrical field was also shown to be sensitive to the twist angle (Fig. R3e, f).

As also indicated in our previous DFT calculations (*ACS Nano* **11**, 6382–6388 (2017)) for a graphene/hBN heterobilayer, the vertical polarizations for the AB and BA stacking configurations are respectively 1.5 pC/m and -0.33 pC/m, so the change in polarization after electric switching will be 1.83 pC/m, very close to that of MoS₂ bilayer (0.97 pC/m - (-0.97 pC/m) = 1.94 pC/m).

Figure R2. The energy difference between AB and BA stacking under electric fields. The red line represents the downward electric field, and the blue line represents the upward electric field. Right insets: Schematics for the two polarized states in real space, where within a moiré supercell, blue (AB configuration) and red (BA configuration) regions are either enlarged or squeezed due to the positive/negative energy differences.

Figure R3. Similar calculation results for twisted bilayer MoS_2 (Reprinted from *npj 2D Mater. Appl.* **6**, 7 (2022)). With (b, d) and without (a, c) electric fields, two polarized configurations (AB and BA) show distinct proportions in area. The polarization and susceptibility also depend on twist angles (e, f).

● DFT Calculation Details

The density functional theory (DFT) calculation was carried out using the projector augmented wave (PAW) (*Phys. Rev. B.* **50**, 17953 (1994)) scheme with the Perdew–Burke–Ernzerhof (PBE) functional of generalized gradient approximation (GGA) (*Phys.*

Rev. Lett. **77**, 3865 (1996)) method as implemented in the Vienna *ab initio* simulation package (VASP) (*Phys. Rev. B.* **54**, 11169 (1996); *Comp. Mater. Sci.* **6**, 15 (1996)). A plane wave cutoff of 750 eV was set in our calculations. K-point samplings of $12 \times 12 \times 1$ was used. optB88 level (*J. Phys.: Cond. Matter* **22**, 022201 (2010)) was used in our calculations to taking into consideration of the van der Waal forces. Atomic relaxation was performed until the force on each atom is smaller than $0.001 \text{ eV \AA}^{-1}$, and the total energy change was less than 10^{-6} eV . The vacuum spacing is set as at least 16 \AA along the out-of-plane direction that is sufficiently large. The electric field action is equivalent to exerting equal and opposite displacements on the oppositely charged atoms carrying Born effective charges in the unit cell (*Phys. Rev. Lett.* **125**, 197601 (2020)), and in this work they are B and N atoms.

Overall, the previously proposed behavior of two polarized states in electric fields are confirmed by calculation, supporting the scenario of enlarging/squeezing areas of states with a lower/higher energy. However, we must stress that *the nonzero net polarization is only valid in perpendicular electric fields*. Once the electric field goes to zero, the net polarization will diminish accordingly, in contrast to the definition of ferroelectricity where a polarized state survives without the presence of an electrical field. Defects, mislocations or strain disorders may preserve the polarization at zero electric fields, but such extrinsic factors are uncontrollable and highly depend on local configurations of a specific device, which are not in line with the uniform and repeatable ferroelectricity in experiments. *Despite these drawbacks, this mechanism was not directly excluded in the manuscript*. This is because it takes place at the interface between graphene and hBN flakes, in complementary to other mechanisms occurring inside the bilayer graphene or the hBN flake. We prefer to keep the possibility of all available mechanisms until more conclusive experimental results are observed.

In response, discussions about the mechanism are revised in lines 162-171, page 7 of the revised manuscript. The above DFT calculations are also added as paragraphs 2-5 in pages 6-7 and Fig. S21 in page 31 of Supplementary Information. All modifications are highlighted in red.

A clear understanding of the mechanism is essential to improve the quality of the present work, and refresh the understanding for the intriguing phenomena.

Reply: We agree with the referee that it is important to unveil the physical mechanism. *The aim of this basically experimental work is to provide essential features and details for constructing a theoretical model*. Our results have clarified several important facts that were misunderstood previously. It can be inferred that (1) electronic correlation and (2) a long-wavelength moiré superlattice may not be the key ingredients, which is believed to inspire more experimentalists

to search for the critical factor of the gate-specific anomalous screening and ferroelectricity. At present – unfortunately – after analyzing all these results including ours and previous reports, a complete understanding could not be achieved.

Actually, the failure of theories is not surprising, because many *fundamental (but unclarified so far)* details have not been captured by the models. For instance, the above theoretical model and others in literatures are (always) based on a structure of crystallographically aligned graphene and hBN. However, in experiments a graphene with a small twisted angle (<5 degrees) on one side and 30 degrees on the other side is found to have a larger probability to observe ferroelectricity, as shown in the discussion in Section 3 of Supplementary Information. Why does the angle of ~30 degrees in the configuration matter, as it is always treated as a smeared background in theories? Are there any other configurations? These basic questions are still open at present. A convincing theoretical scenario is not possible until such kinds of details can be provided by experiments. We hope the present work could stimulate smoking-gun experiments in the future to figure out the accurate range of twisted angles.

Reviewer #2 (Remarks to the Author):

I read through the revised manuscript and the response letter. The authors fully addressed my comments. I thus recommend a publication.

Reply: We thank the referee for the recommendation.

Reviewer #3 (Remarks to the Author):

In my first report I recommended this paper for publication in Nat. Comms., only requesting that the authors commented on the microscopic origin of the ferroelectricity that they measure in the graphene-hBN heterostructures described in their manuscript. As a response, the authors have supplemented their manuscript with a detailed discussion of possible mechanisms behind this ferroelectric effect. I'm satisfied with this and my recommendation to publish the paper stands.

Reply: We thank the referee for the recommendation.

REVIEWERS' COMMENTS

Reviewer #1 (Remarks to the Author):

In the second round revision, the authors solve the problems of theoretical simulation to assess the polarization strength and the energy difference under the vertical external field, which is helpful to understand the physics essence to some degree. However the polarization will disappear after the electric field is removed, which actually don't meet the definition of ferroelectricity. Although defects, strain gradient and mislocations can probably preserve the polarization as argued, they are more possibly harmful for the stability and existence of ferroelectric domains. The lack of convincing experimental evidences will render the argument a bit weak.

Even so, it is understandable that the comprehensive understanding for the twisted induced ferroelectricity is extremely challenging due to the facts that: 1, it is hard for experiments to detect the atomic details of stacking and rotations. 2) Current theories can't consider very complex structures. This work is interesting and may simulate the further investigations from the communities even the underlying mechanism is not clear in the work.

Overall, with all the factors in considerations, I will not oppose its publication in Nature Communications.

Reviewer #1 (Remarks to the Author):

In the second round revision, the authors solve the problems of theoretical simulation to assess the polarization strength and the energy difference under the vertical external field, which is helpful to understand the physics essence to some degree. However the polarization will disappear after the electric field is removed, which actually don't meet the definition of ferroelectricity. Although defects, strain gradient and mislocations can probably preserve the polarization as argued, they are more possibly harmful for the stability and existence of ferroelectric domains. The lack of convincing experimental evidences will render the argument a bit weak.

Even so, it is understandable that the comprehensive understanding for the twisted induced ferroelectricity is extremely challenging due to the facts that: 1, it is hard for experiments to detect the atomic details of stacking and rotations. 2) Current theories can't consider very complex structures. This work is interesting and may simulate the further investigations from the communities even the underlying mechanism is not clear in the work.

Overall, with all the factors in considerations, I will not oppose its publication in Nature Communications.

Reply: We thank the referee for positive comments and encouragement. The mismatch between experiments and theoretical models is believed to motivate deeper and more extensive investigations in the future.